# Community Battery for Collective Self-Consumption and Energy Arbitrage: Independence Growth vs. Investment Cost-Effectiveness

**Mattia Pasqui** [1,*], **Lorenzo Becchi** [2], **Marco Bindi** [2] , **Matteo Intravaia** [2] , **Francesco Grasso** [2] , **Gianluigi Fioriti** [2] and **Carlo Carcasci** [1]

1   Department of Industrial Engineering (DIEF), University of Florence (UNIFI), 50121 Florence, Italy; carlo.carcasci@unifi.it
2   Department of Information Engineering (DINFO), University of Florence (UNIFI), 50121 Florence, Italy; lorenzo.becchi@unifi.it (L.B.); m.bindi@unifi.it (M.B.); matteo.intravaia@unifi.it (M.I.); francesco.grasso@unifi.it (F.G.); gianluigi.fioriti@unifi.it (G.F.)
*   Correspondence: mattia.pasqui@unifi.it; Tel.: +39-3342798301

**Abstract:** Integrating a grid-connected battery into a renewable energy community amplifies the collective self-consumption of photovoltaic energy and facilitates energy arbitrage in the electricity markets. However, how much can energy independence really increase? Is it a cost-effective investment? The answer to these questions represents a novelty in the literature due to the innovative nature of the asset under consideration and the market and regulatory framework in which it is evaluated. Employing a net present value assessment, our analysis incorporated aging effects and conducts sensitivity analyses across various parameters: the number of community customers, electricity market prices, battery cost and size, and the decision to engage in energy arbitrage. Each scenario underwent a 20-year hourly simulation using an aging-aware rolling-horizon 24 h-looking-ahead scheduling, optimized with mixed-integer linear programming. Simulations conducted on the Italian market indicate that dedicating a battery solely to collective self-consumption is the most efficient solution for promoting a community's energy independence, but it lacks economic appeal. However, integrating energy arbitrage, despite slight compromises in self-sufficiency and battery longevity, halves the payback period and enhances the attractiveness of larger battery investments. The net present value is contingent upon the battery size, customer number, and market prices. Nevertheless, if the battery cost does not exceed 200 EUR/kWh, the investment becomes cost-effective across all scenarios.

**Keywords:** renewable energy community; battery energy storage system; scheduling; aging; collective self-consumption; energy arbitrage

## 1. Introduction

The electricity system is undergoing a shift from centralized to decentralized production. However, integrating decentralized renewable energy systems into the grid poses challenges due to resource intermittency. Battery Energy Storage Systems (BESS) and renewable energy communities (RECs) play crucial roles in tackling these challenges.

European directives incentivize active consumer participation in renewable energy production. RECs promote aggregation for energy production, consumption, storage, and sharing, often leveraging photovoltaic systems. Collective self-consumption (CSC), incentivized monetarily, encourages consumers to align usage with production, enhancing incentives and grid independence. REC not only focuses on CSC incentives but also seeks to engage in electricity markets. This study examines conditions for REC investment in a community battery to enhance CSC and participate in day-ahead and intra-day electricity markets through energy arbitrage (EA), aiming to profit by buying low and selling high.

### 1.1. Legislative Framework and Literature Review

The evolving EU energy policies, guided by regulations and directives, prioritize reducing greenhouse gas emissions and enhancing energy efficiency by increasing renewable energy sources (RES) in electricity production [1]. This transition toward RES integration in the grid focuses on two strategies: enhancing demand flexibility through Battery Energy Storage Systems (BESS) and fostering renewable energy communities (RECs) [2]. Directives 2019/944 [3] and 2018/2001 [4] facilitate citizen engagement with the electricity system, encouraging active involvement and aggregation within RECs. These communities engage in various market activities, including generation, consumption, sharing, trading, and providing flexibility services via demand response and energy storage. Member States, in compliance with these directives, are devising mechanisms to enable consumer participation in energy communities, offering incentives to expedite their deployment.

Italy has implemented REC legislation through specific legal provisions and regulations [5–9]. In Italy, a REC constitutes a virtual community where consumers and producers collectively produce, consume, store, and share energy from renewable sources. Energy sharing, termed collective self-consumption (CSC), is incentivized by the Italian government at approximately EUR 110/MWh and can be utilized for community activities or redistributed among members. To optimize CSC, a grid-connected BESS can be utilized [8,9], whereby energy withdrawn for subsequent feed-in is added to the collective self-consumption calculation. Participation in electricity markets is necessary for BESS to exchange energy with the grid, with operational details governed by Italian regulators [10,11].

There is a noticeable gap in the literature concerning Battery Energy Storage Systems (BESS) within renewable energy communities (RECs). While interdisciplinary literature on RECs is growing, it often overlooks the specific role of BESS within RECs. Conversely, extensive research exists on utility-scale grid-connected BESS providing multiple services, yet its application to RECs remains unexplored.

RECs across Europe vary due to factors such as energy technology, sources, and regional regulations [12,13]. They can be physically or virtually configured, with only the virtual option permitted in Italy, as it utilizes the national grid [14,15]. This study focuses on the virtual configuration, employing energy from photovoltaic panels or the national grid. Economic aspects dominate REC literature, comparing various business models and addressing incentive redistribution and cost allocation [16–21]. Some explore peer-to-peer trading, demand-side management, and REC composition and configuration [22–27]. The primary research question concerns the economic conditions necessary for REC viability and how stakeholders contribute to community sustainability [19,20,27,28]. While economic evaluations in these studies rely on energy simulations, few delve into the role of the battery. Some assess battery sizing's influence on self-consumption and REC gains [29,30]. Others investigate the battery's impact on the distribution grid, scheduling processes, and the possibility of aggregating multiple batteries or even heat pumps [31–36]. However, the use of grid-connected BESS within RECs remains underexplored. Contributions in this field propose community BESS for energy arbitrage and peak shaving [37]. However, these perspectives focus on Distribution System Operators (DSOs) rather than RECs and do not integrate collective self-consumption into scheduling algorithms. Moreover, the literature lacks assessments of investment costs and battery aging [38].

A review highlights the need for community BESS to cater to multiple services to optimize utility and economic gains [39]. However, the literature on BESS providing multiple services primarily focuses on utility-scale applications rather than RECs. These services can be classified into four mainstream categories:

- The provision of ancillary services (AS) to the grid operator to enhance the system reliability (e.g., frequency containment, frequency restoration, and replacement reserve).
- Dispatching, i.e., real-time coverage of dispatching errors.
- The achievement of local objectives, such as self-consumption and collective-self-consumption (CSC).
- Energy arbitrage (EA), i.e., buying and selling electricity to generate revenue.

Models in the literature mainly focus on AS and dispatching. Ref. [40] proposes a general framework for the scheduling and control of a BESS to provide multiple services and uses it in the problem of providing dispatchability. This problem is explored in more detail in [41], adding grid constraints and proposing a two-level control layer to avoid battery saturation. In [42], the provision of AS is also added to the problem's formulation. For a review of the possibility of providing AS using BESS focused on Italy's market and regulation, see [43]: a market price sensitivity analysis compared to the economic feasibility of the investment is performed. Instead, the BESS modeling methodology for stacking more than one ancillary service is described in [44]. A more comprehensive overview of how a BESS can provide multiple services and the programming methodologies used in various cases is beyond the scope of this article (see [45]).

Meanwhile, the use of a community battery for multiple services is overlooked, and scheduling algorithms for multiple services have not yet been applied to CSC and EA in REC literature.

### 1.2. Novelties

In this context, this paper's contributions can be summarized as follows:

- It introduces a new aging-aware rolling-horizon model for the hourly scheduling of a community battery. While existing battery scheduling models cover multiple services, integrating CSC and EA into these models is a novel addition. This novelty stems from the recent emergence of both CSC and EA concepts. The former is obviously related to the new appearance of RECs. The latter has only recently become feasible with the development of the intraday market, allowing bidding up to an hour before delivery based on reliable forecasts and knowledge of the day-ahead market prices.
- It conducts an extensive sensitivity analysis on various scenarios to explore the economic feasibility of investing in a community battery. Five key parameters are considered: community size, electricity market prices, battery cost, size, and the decision to engage in energy arbitrage. Such a comprehensive techno-economic analysis of this asset has not yet been proposed in the literature on RECs.
- Additionally, the scheduling model takes into account battery aging, as does the investment assessment. The combined effects of the provision of EA and CSC services on aging have not been previously studied.

### 1.3. Limitations

The primary limitations are as follows:

- Forecast errors are not considered. Indeed, scheduling assumes deterministic knowledge of future load and production. However, considering that scheduling is a rolling horizon and takes place one hour before delivery, i.e., at the close of the intraday market, forecast errors should be limited.
- Real-time control is not implemented, and at the same time, the costs of imbalances are not included in the economic calculation. This point is complicit with the previous assumption because if the forecasts are perfect, there are no imbalances and no need for a control to reduce them, performing dispatching.
- Simplified participation in the day-ahead and intraday markets is assumed, where all bids can be submitted at the closure of the latter market without differences in prices between the two markets. However, in reality, initial scheduling should occur at the closure of the day-ahead market, followed by continuous rescheduling during the intraday market as the delivery time approaches. The cost of rescheduling due to price differences between the two markets, albeit low in the Italian context, is not included in the economic evaluation.
- The provision of ancillary services in the balancing market is not evaluated, but it could certainly serve as an additional revenue stream for a community battery.

- The electricity grid is not modeled, which is definitely an aspect to consider to fully complete evaluations like those proposed. Scheduling without considering grid constraints could lead to bidding solutions that are technically undeliverable.

These limitations foreshadow future articles and the direction for further developments in broader research, positioning this study as an initial building block.

## 2. Materials and Methods

The chapter begins by introducing the selected case study for simulations, detailing the concept of a renewable energy community (REC) within Italian regulatory frameworks. It explains how a Battery Energy Storage System (BESS) can actively engage in collective self-consumption (CSC) and energy arbitrage (EA).

After the case study introduction, the BESS model is elaborated upon. A mixed-integer linear programming (MILP) approach is utilized to compute BESS scheduling, considering relevant parameters and techniques to address battery aging effects.

Next, the economic evaluation formulations are presented to assess the financial feasibility of the proposed system. Finally, simulated scenarios are introduced, followed by a comparative analysis of these scenarios in the Results chapter.

### 2.1. Case Study

A renewable energy community (REC) fueled by photovoltaic systems with an overall power of 100 kWp is examined, exclusively comprising residential customers. Photovoltaic power is the REC reference size and is kept constant during simulations. However, the results are scalable to RECs with higher production. On the other hand, battery size and number of consumers are the subject of sensitivity studies. According to the Italian regulation, collective self-consumption (CSC), which is the virtual self-consumption of the whole community, is an incentive at about 110 EUR/MWh. It is specifically defined as the minimum on an hourly basis between the feeding and withdrawal by all members of the REC. The energy withdrawn from a grid-connected Battery Energy System (BESS) for the purpose of subsequent feed-in (green row in Figure 1) is added to the energy withdrawn to calculate the CSC. This is why this article assesses using a BESS to increase the CSC and thus the incentive.

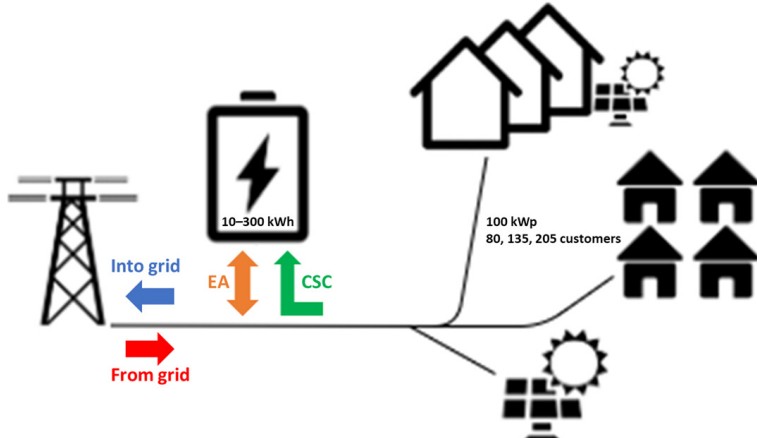

**Figure 1.** Case study definition.

In order to be able to exchange energy with the grid, the BESS must participate in electricity markets. In particular, this article considers participation in the day-ahead and intraday markets. This introduces the possibility of performing energy arbitrage (EA) by buying energy when prices are low and reselling it when they are high.

### 2.2. BESS Scheduling Model

According to the definition of CSC and the electricity market, the simulations performed have an hourly time step. However, in the near future, the market will become quarter-hourly. Considering that the intraday market closes an hour before delivery [46], the optimal battery scheduling for the next 24 h is calculated every hour, and the first hour is used as simulation. The scheduling problem is defined as a mixed-integer linear programming (MILP) optimization model. This problem is rolling horizon, because it is solved each hour, and it looks 24 h ahead. Such problem is solved for each hour of the yeas (8760 h) for 20 years, so each simulation is a combination of 8760 per 20 MILP optimizations. A deterministic knowledge of production, load, and energy price for the next 24 h is considered. These series serve as inputs for the MILP problems.

A thorough description of the scheduling optimization model follows objective functions (Equations (1)–(4)) and constraints (Equations (5)–(8)).

$$f_{obj}(E_{bess}) = \sum_{h=1}^{24} EA_h + CSC_h - AP_h \quad [\text{€}] \tag{1}$$

$$EA_h = E_{bess,h} \cdot EP_h \tag{2}$$

$$CSC_h = \min[E_{sur,h}, \max(0, E_{bess,h})] \cdot \text{inc} \tag{3}$$

$$AP_h = \frac{|E_{bess,h}| \cdot AC}{2} \tag{4}$$

The objective function $f_{obj}$ is an economic one. It is the sum on 24 h of the revenue obtained from energy arbitrage (EA) and collective self-consumption (CSC) minus an Activation Penalty (AP), which is linked with BESS aging and replacement cost. Aim of the optimization problem is to maximize the objective function.

The hourly energy BESS exchanges with the grid ($E_{bess}$) is the variable to be optimized: a vector of length 24 representing the scheduling of the battery. $E_{bess,h}$ is negative if the battery feeds energy into the grid, or positive if the battery draws energy.

$EP_h$ represents the hourly Energy Price. The price is always negative, so if energy is withdrawn, the product $E_{bess,h} \cdot EP_h$ is a cost, while when it is fed, it is a gain.

The gain for CSC is the product between the value of the incentive (inc = 110 EUR/MWh) and the energy drawn from BESS that is counted as CSC. The latter is the minimum between the REC energy surplus ($E_{sur,h}$) and the energy drawn by BESS, i.e., the positive values of $E_{bess,h}$ ($\max(0, E_{bess,h})$).

The penalty due to activation linked with aging is the product of the amount of energy fed or withdrawn ($|E_{bess,h}|$) and the Activation Cost (AC) parameter, whose function will be explained in the next subsession.

The constrains of the model are the following:

$$SoC_{h+1} = SoC_h + E_{bess,h} \cdot \eta \tag{5}$$

Define the State of Charge (SoC) variable, which is dependent on parameter $E_{bess}$ and considers an average charging and discharging efficiency ($\eta$) of 0.90 [47].

$$C_{max} \cdot DoD \leq SoC_h \leq C_{max} \tag{6}$$

$$|E_{bess,h}| \leq C_{max} \tag{7}$$

SoC and $E_{bess}$ box constrains consider BESS maximum capacity ($C_{max}$) and depth of discharge (DoD). DoD is fixed to 0.10, while $C_{max}$ is equal to size of BESS at the beginning of each simulation but then decrease due to aging effects. Therefore, it is assumed in Equation (7) that the battery can be fully charged or discharged in one hour, with an average q-rate = 1.

$$-E_{need} \leq E_{bess,h} \leq E_{sur} \tag{8}$$

This constraint obliges the battery to only be able to charge with the energy surplus of REC ($E_{sur}$) and to only discharge to meet the REC's energy need ($E_{need}$). This constraint is aimed at preventing EA through the purchase or sale of energy from outside the REC, which is not necessary for CSC. The constraint is active in scenarios with only CSC and inactive when EA is also desired.

There are also additional constraints and dummy variables in the model that serve for the linearization of the functions absolute value, minimum, and maximum.

### 2.3. BESS Aging Awareness

Figure 2 explains the effect of Activation Cost (AC) added to the MILP model (see Equation (4)) and its connection to BESS aging. Essentially, the AC value represents the minimum price difference required for executing a charge and discharge cycle to be advantageous. The division by 2 in Equation (4) precisely aligns the AC value with the buy-and-sell price difference. A high AC value corresponds to a low number of cycles, and vice versa (Figure 2c). However, a low number of cycles results in lower earnings in EA (Figure 2a). These effects can be evaluated over several years considering the Net Present Value (NPV) evolution (Figure 2b). Year after year, the available battery capacity decreases, and so does the cash flow. When end of life is reached, the battery needs to be replaced (decline steps in Figure 2b). As explained in the next paragraph, in the calculation of the NPV, a replacement is included as an additional cost that is subtracted from the regular cash. For this reason, in the year of replacement, the cash flow is negative, and in fact, there are "decline-steps" in the NPV graph in corresponding to the years when replacement is necessary. With high AC values, the battery ages more rapidly, but annual earnings are higher; with low AC values, the battery lasts longer, but the earnings are lower.

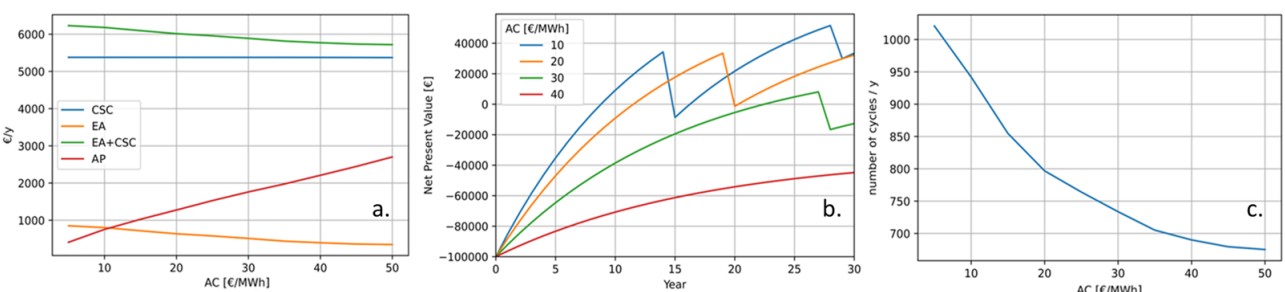

**Figure 2.** Effects of Activation Cost (AC) on collective self-consumption (CSC) and energy arbitrage (EA) cash flow (**a**), on Net Present Value (NPV, (**b**)), and on number of cycles per year (**c**).

To calculate battery aging, a rain flow counting method [48] has been used to calculate the equivalent number of cycles undergone by the BESS. Assuming that the BESS reaches its end of life after 8000 cycles with 80% remaining capacity [49], the available capacity is recalculated weekly in proportion to the number of equivalent cycles reached. Upon reaching 8000 cycles, the BESS is replaced (decline steps in Figure 2b). This empirical and macroscopic approach to calculating aging is considered sufficient for the purposes of this paper. While using equivalent circuit models or physical (i.e., electrochemical) models could provide more precise estimates of aging, they are difficult to generalize to different storage technologies and require higher computational costs. The proposed approach, however, is simple to implement; one only needs to write the rain flow counting algorithm and enter the end-of-life information on the battery, which is easily obtainable from any manufacturer. Although the aging estimate may be rough, it is sufficient for the purposes of this paper.

*2.4. Economic Analysis*

The economic assessment of BESS investment is based on the Net Present Value (NPV), calculate for 20 years (y) using Equations (9)–(11).

$$NPV_y = NPV_{y-1} + \frac{CF_y}{(1+i)^y} \tag{9}$$

$$CF_y = EA_y + CSC_y - Cost_{repl} \cdot R \tag{10}$$

$$NPV_0 = Size_{bess} \cdot Cost_{bess} = Cost_{repl} \tag{11}$$

$NPV_0$ represents the initial investment for BESS, and $CF_y$ denotes the annual cash flow, encompassing the sum gains from EA and CSC. The annual interest rate, denoted as i, is set at 5%. When BESS replacement occurs (R = 1; otherwise, R = 0), replacement cost ($Cost_{repl}$) is incorporated into $CF_y \cdot Cost_{repl}$ is assumed to be equal to the initial installation cost ($NPV_0$), which is equal to $Size_{bess}$ per $Cost_{bess}$.

To facilitate the comparison of diverse investments with replacements occurring in different years, a transformation of NPV is employed in this study. Looking at the left-hand image in Figure 3, it is difficult to identify the value of the activation cost (AC) that determines the optimal NPV, because the choice depends on the specific year in which the NPV is compared. However, using the transform shown in the image on the right clarifies the most favorable AC value (i.e., ac = 35).

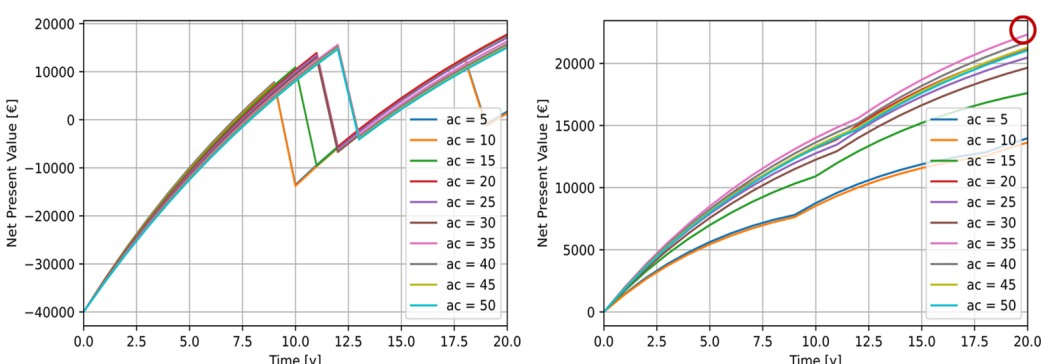

**Figure 3.** Effects of Net Present Value (NPV) (**left**) and NPV transformation (**right**) used to compare different Activation Costs (ACs).

The transformation used for NPV can be explained by observing Figure 4. The investment presented by the NPV line in blue can be transformed into the equivalent investment of the orange line, which does not have discontinuity due to the lack of replacement of the initial investment, which is equal to zero. To compute the transformation, it is essential to consider that both the initial investment cost and the replacement costs are not paid upfront. Instead, they are financed through a loan with a duration equivalent to the battery's lifespan and a loan interest rate chosen so that the original NPV and the transformed NPV are equal in the years when replacements occur. Thus, the transformed Cash Flow (CF*) for the transformed NPV must be recalculated with respect such conditions. Equations (12)–(15) synthesize how the NPV transformation can be calculated, where LF is the Loan Factor to be considered in the transformed cash flow.

$$NPV_y^* = NPV_{y-1}^* + \frac{CF_y^*}{(1+i)^y} \tag{12}$$

$$CF_y^* = EA_y + CSC_y - LF \tag{13}$$

$$LF = \frac{Size_{bess} \cdot Cost_{bess}}{\sum_{y=0}^{lifespam_{bess}} \frac{1}{(1+i)^y}} \tag{14}$$

$$NPV_0 = Cost_{repl} = 0 \tag{15}$$

Figure 3 shows that the variation of NPV with AC values is not strictly monotonic but exhibits a global maximum (35 in the example), alongside several local peaks. This peculiar trend primarily stems from the non-uniform distribution of energy prices in the electricity market, which, contingent upon the AC value, impacts both cash flow and battery aging, consequently exerting further influence on cash flow. The outcome of such intricate interplay can only be computed through simulations employing a detailed time step and spanning a lengthy time horizon, such as those proposed.

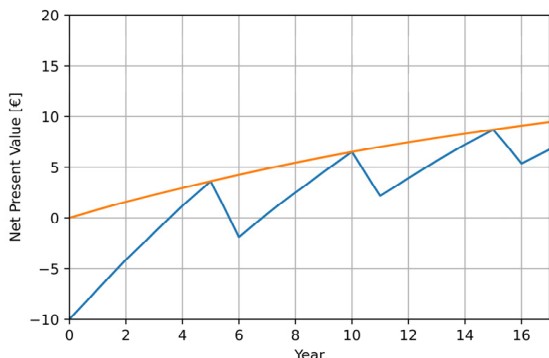

**Figure 4.** Net Present Value (NPV) (blue line) and NPV transformation (orange line).

### 2.5. Simulated Scenarios

This study encompasses the simulation of 36 distinct scenarios aimed at evaluating the impact of 4 key parameters (Table 1): number of customers in the renewable energy community (REC), energy price, battery cost, and the possibility of performing energy arbitrage (EA). The first parameter pertains to the REC energy surplus ($E_{sur}$) and needs ($E_{need}$), which the BESS can harness to derive gains through CSC. The second parameter influences earnings from EA. Battery cost ($Cost_{bess}$) has an impact on initial investment and replacement cost, and thus on NPV. Performing EA influences cash flow.

**Table 1.** Simulated scenarios.

| Parameter | Scenarios |
|---|---|
| Customer number (CN) | 80, 135, 205 residential customers |
| Energy price (EP) | Low and high prices (2020 and 2023) |
| Battery cost ($Cost_{bess}$) | 200, 400, 600 EUR/kWh |
| Energy arbitrage (EA) | CSC + EA vs. CSC |

Within each scenario, a substantial number of simulations are conducted to perform sensitivity analysis on two primary variables (Table 2): $Size_{bess}$ and AC. Therefore, for every combination of scenarios, $Size_{bess}$ values, and AC settings, a simulation spanning 20 years is executed. Within this simulation, each hour is an outcome of a MILP optimization process.

**Table 2.** Sensitivity analysis.

| Variable | Range |
|---|---|
| Battery size ($Size_{bess}$) | 20 to 300 kWh |
| Activation cost (AC) | 5 to 60 EUR/MWh |

Energy Price (EP) is an array comprising 8760 values, corresponding to the number of hours in a year, and is repeated over 20 years. Also, the vectors representing REC energy surplus ($E_{sur}$) and need ($E_{need}$), which are calculated depending on customer number (CN), have the same dimension. What evolves annually is the available capacity of the BESS, which diminishes due to aging. The procedure for selecting the two EP scenarios and the three CN scenarios (from which $E_{sur}$ and $E_{need}$ are dependent) is outlined below.

Considering the information about Energy Price (EP) in the Italian electricity market reported in Figures 5 and 6 [46], two different scenarios are selected. As a low-price scenario, the EP of 2020 is chosen as the worst-case scenario, and as a high-price scenario, 2023 prices are considered. The years 2021 and 2022 were excluded due to their excessive anomalies and randomness.

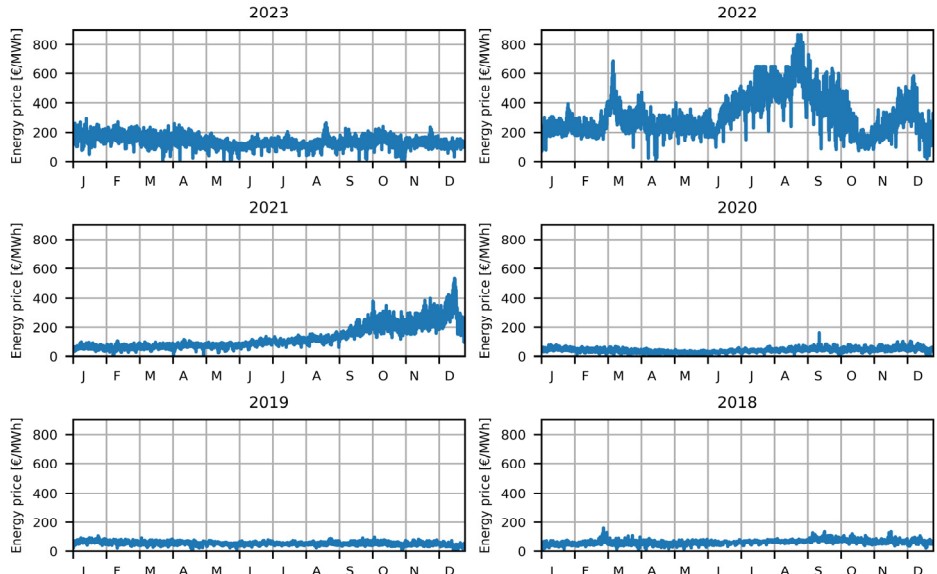

**Figure 5.** Hourly energy price in the Italian electricity market from 2018 to 2023.

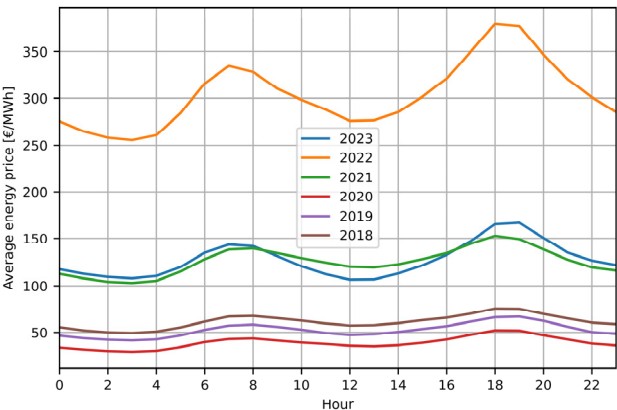

**Figure 6.** Average hourly energy price in Italian market from 2018 to 2023.

Dataset used for photovoltaic production downloaded from PVgis [50] are also from 2020 and 2023. This correspondence ensured that the price trends remained consistent with the fluctuations in production. The input parameters for PVgis included the geographical coordinates of Florence, a tilt angle of 30°, an azimuth angle of 0°, and losses of 14%.

To calculate the $E_{sur}$ and $E_{need}$ of the REC, in addition to the photovoltaic production series, the aggregated consumption series of all community customers is required. A load series representing a 3 kWp typical residential consumers from Tuscany is generated by utilizing average hourly profiles provided by the Italian regulatory authority [51] (Figure 7).

These profiles are differentiated based on the month and the type of day. The resulting load series is subsequently scaled by the number of customers within the considered REC, aggregating the total consumption pattern.

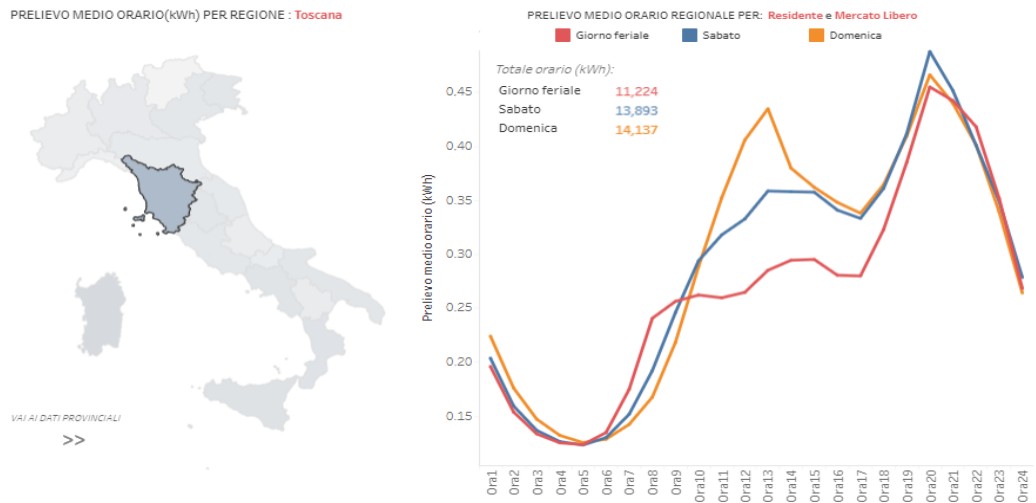

**Figure 7.** Average load profiles in January for a 3 kW residential customer in Tuscany.

Using the open-source multi-energy simulation software "MESSpy" [52,53], a sensitivity analysis was conducted on varying the number of customers within the REC (Figure 8). Based on this analysis, three scenarios were selected (Table 3) to be simulated with the BESS, representing collective self-consumption indices of 40%, 60%, and 80%.

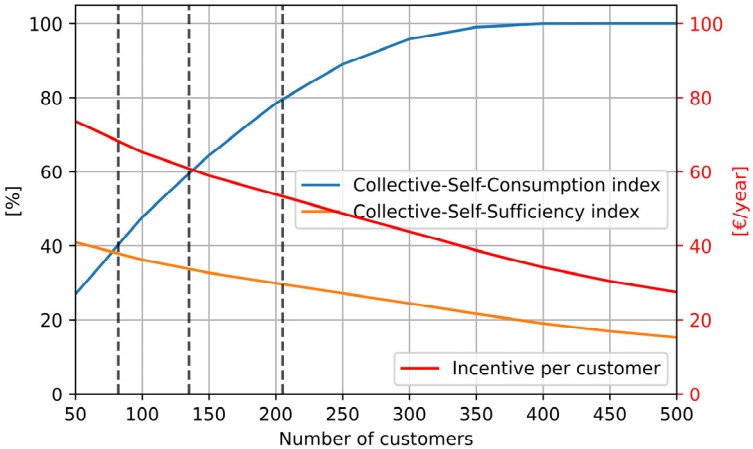

**Figure 8.** The 100 kWp REC key parameters, with varying numbers of customers.

**Table 3.** Three scenarios selected with regard to customer number.

| Number of Customers [CN] | CSCi [%] | CSSi [%] | E$_{sur}$ [MWh/year] | E$_{need}$ [MWh/year] |
|---|---|---|---|---|
| 80 | 40 | 38 | 82 | 89 |
| 135 | 60 | 34 | 55 | 160 |
| 205 | 80 | 30 | 27 | 258 |

Considering that collective self-consumption (CSC) is defined as the minimum between the energy injected into the grid by the community and the energy withdrawn (including that withdrawn from the battery), we consequently defined the following two relative indices: the collective self-consumption index (CSCi) is defined as the ratio of

CSC to the total electricity production from photovoltaic sources, while the collective self-sufficiency index (CSCi) is the ratio of CSC to the total energy demand of the customers. These two indexes are representative of the REC's independence from the national grid. On the other hand, the incentive associated with CSC is the multiplication of CSC by a value of approximately 110 EUR/MWh.

Figure 8 not only identifies three potential scenarios but also elucidates why evaluating a battery within future RECs is sensible: as the number of customers increases, so does the demand. With the withdrawal of electricity, the CSC, the total incentive, and CSCi rise while the CSSi diminishes. Moreover, the incentive per customer decreases. In practical terms, with more users, the metaphorical "pie" must be divided into more portions, leading to smaller individual slices. This observation highlights the impracticality of considering RECs with excessively high CSCi levels, suggesting that surplus energy ($E_{sur}$) will persist in future RECs. This raises the question of who will harness this surplus if not through grid-connected community batteries. Thus, the graph underscores that RECs will consistently have surplus energy, justifying the evaluation of introducing a BESS to harness and capitalize on this surplus.

## 3. Results

The outcomes are delineated across three segments. Firstly, we elucidated the impact of the Battery Energy Storage System (BESS) on renewable energy community (REC) energy balances. Secondly, we delineated the significance of the activation cost (AC) parameter with regard to battery degradation and the Net Present Value (NPV). Lastly, we conducted economic optimization of battery sizing within each scenario and appraised potential investments by performing energy arbitrage (EA) or dedicating the BESS only to collective self-consumption (CSC).

### 3.1. Energy Balances

Upon integrating a BESS into a REC, its grid independence increases. The collective self-consumption (CSC) and collective self-sufficiency (CSS) indices, reflecting REC autonomy from the grid, increase with larger battery sizes (Figure 9). However, the degree of increase varies with the performance of EA: the augmentation of CSS is constrained by regulations pertaining to access to incentives via the battery. Notably, while withdrawing energy during surplus periods is incentivized, injection during times of need lacks similar encouragement. Consequently, injections may not necessarily occur during these energy needs, which could otherwise enhance CSS, but rather during periods of elevated pricing. Conversely, in the absence of EA, the battery is limited to utilizing surplus energy from the REC and injecting it when required, thereby aligning the enhancement of CSC and CSS in this scenario.

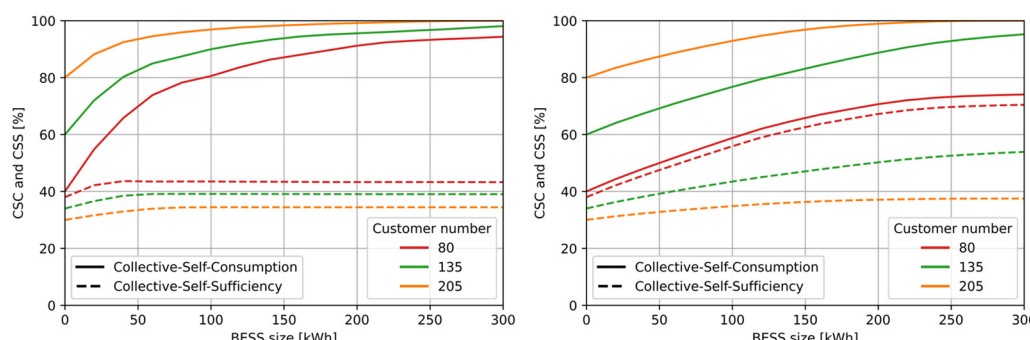

**Figure 9.** CSC + EA (**left**) vs. CSC (**right**): collective self-consumption index and collective self-sufficiency index with regard to varying battery size and number of customers.

Figures 10 and 11 underscore a key revelation from this study. The REC's reliance on the grid, considering the BESS as part of the REC, paradoxically rises instead of declining.

Over a 20-year span, both the energy fed into and withdrawn from the grid increase due to the battery's independent EA activities, notwithstanding decreases in energy exchange within the REC. To address this, the approach advocated in the right-hand graphs restricts the battery to charge solely from the REC surplus and discharge solely to meet community needs. This strategy of non-EA reduces overall energy fed into and withdrawn from the grid, thus mitigating REC energy dependence.

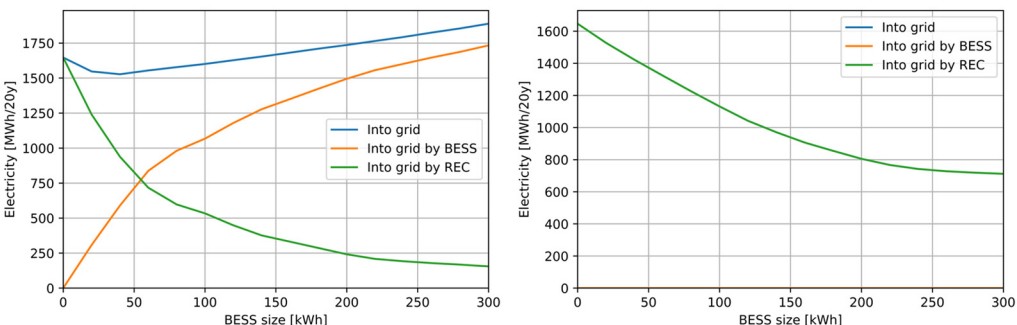

**Figure 10.** CSC + EA (**left**) vs. CSC (**right**): energy fed into the grid with regard to varying BESS size.

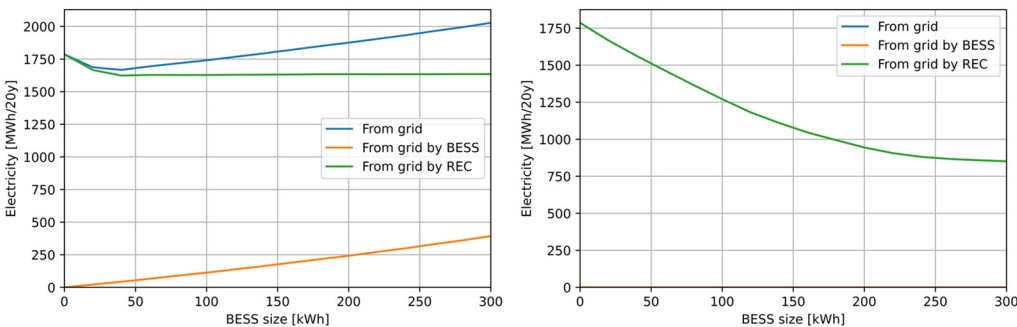

**Figure 11.** CSC + EA (**left**) vs. CSC (**right**): energy withdrawn from the gird with regard to varying BESS size.

### 3.2. Activation Cost and Energy Arbitrage

Adjusting the Activation Cost (AC) parameter across various levels results in the computation of diverse optimal BESS schedules. Over a 20-year analysis period, these variations in AC values significantly impact cash flows and BESS lifetimes, consequently influencing the Net Present Value (NPV), which is highly dependent on AC. Simulations encompassing each scenario and BESS size have been conducted with AC values ranging from 5 to 60.

Figure 12 provides a summary of the optimal AC values, maximizing the NPV transformation (NPV*) over 20 years. These simulations focus on scenarios performing EA. Since AC does not influence the gain from CSC (see Figure 2), there is no point in studying its effects in scenarios without EA. The results are presented with confidence intervals, wherein NPV* values differ by less than 1% of the NPV* value. The battery cost, equivalent to the replacement cost at the end of its life, emerges as the most influential parameter. For BESS costs around 200 EUR/kWh, it is advisable to set AC values between 20 and 30 to enhance EA profits, albeit at the expense of accelerating battery consumption with numerous cycles. Conversely, for higher costs, optimal values shift toward 50, indicating fewer cycles but highly profitable ones, ensuring prolonged battery longevity. The graph also illustrates that the battery size and energy price do not significantly influence AC selection.

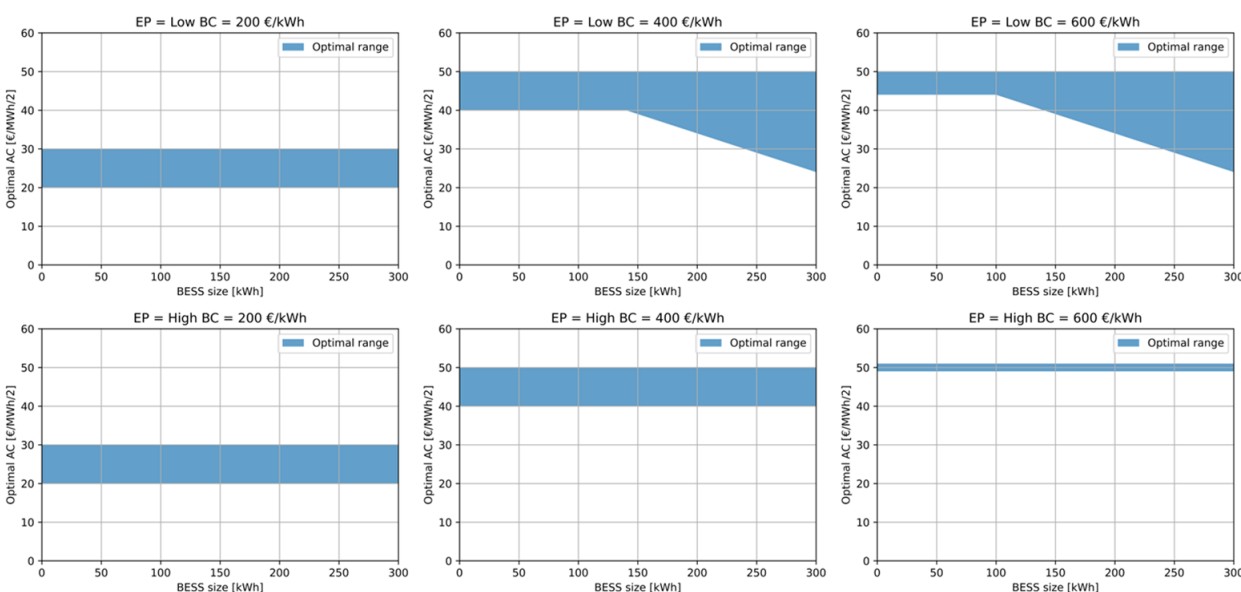

**Figure 12.** CSC + EA: optimal Activation Cost (AC) values for varying Energy Price (EP), battery cost (BC), and battery size.

Summing up, the importance of optimizing battery scheduling considering the expected replacement cost is evident. One way to achieve this is proposed in this study by optimizing the AC parameter entered as a penalty of the objective function.

### 3.3. Economic Feasibility

The findings presented in this concluding paragraph exclusively pertain to configurations with optimal Activation Cost (AC) values. Figures 13 and 14 delineate the respective contributions of the two principal cash flows, energy arbitrage (EA) and collective self-consumption (CSC), to the investment's returns.

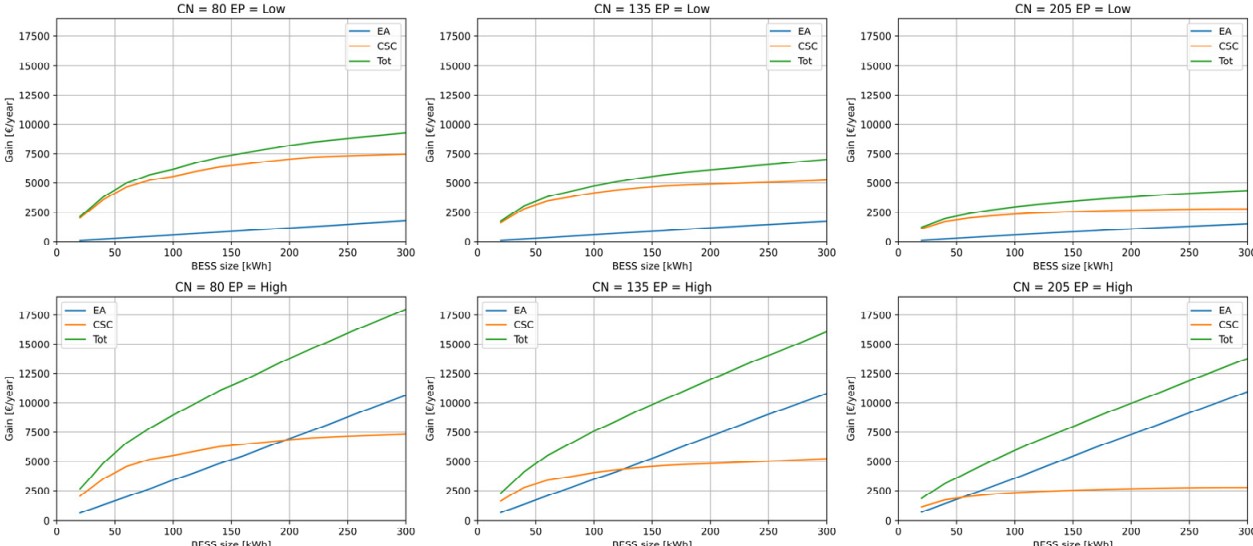

**Figure 13.** CSC + EA: cash flow for varying number of customers (CN), Energy Price (EP), and BESS size.

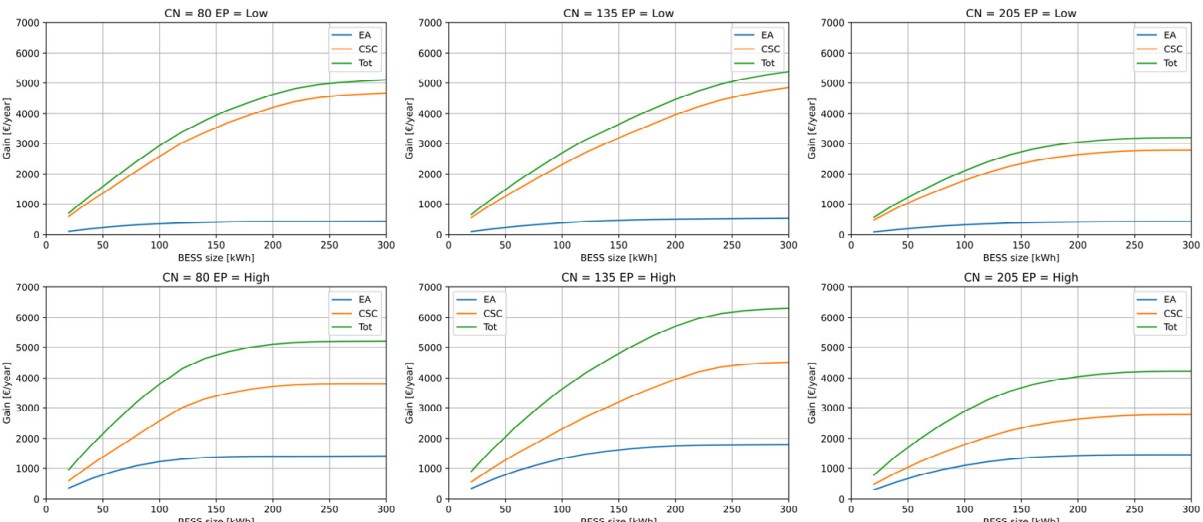

**Figure 14.** CSC: cash flow for varying number of customers (CN), Energy Price (EP), and BESS size.

The analysis reveals that in scenarios featuring EA (Figure 13) and characterized by low energy prices, the gain derived from CSC significantly surpasses that of EA, exceeding it by approximately fivefold. Consequently, investments in such scenarios are primarily driven by REC incentives and are contingent upon the evolution of customer numbers over time. In other words, it is crucial to match the BESS to a REC with a lot of energy surplus. However, with high energy prices, the profit from EA may indeed outstrip that from CSC.

In scenarios without EA use (Figure 14), the EA gain is only due to the buying and selling of energy at times of surplus and need in the REC and not to an actual EA that exploits electricity market price fluctuations. In these cases, the total cash flow experiences a decline by several thousand EUR per year (see *y*-axis scale), chiefly due to reduced EA profits but also due to lower CSC levels. However, if energy prices are high, EA continues to make an important contribution of about one-third of the total cash flow.

Figures 15 and 16 depict the transformed Net Present Value (NPV*) as it relates to BESS size, offering insights into optimal BESS sizes for each scenario of Table 1.

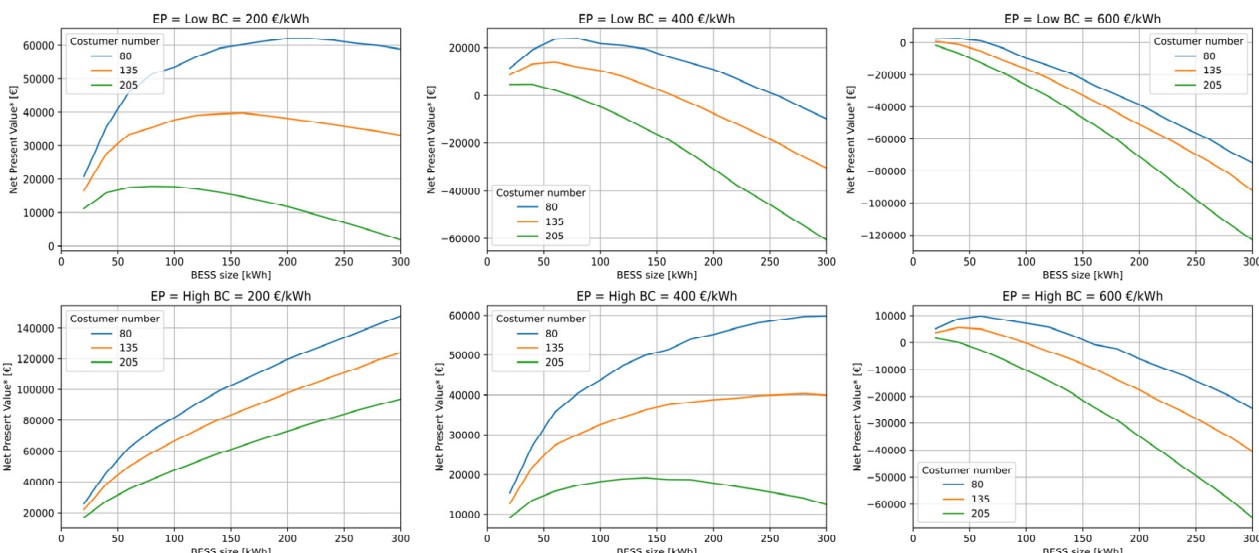

**Figure 15.** CSC + EA: NPV* varying Energy Price (EP), BESS cost (BC), number of customers, and BESS size.

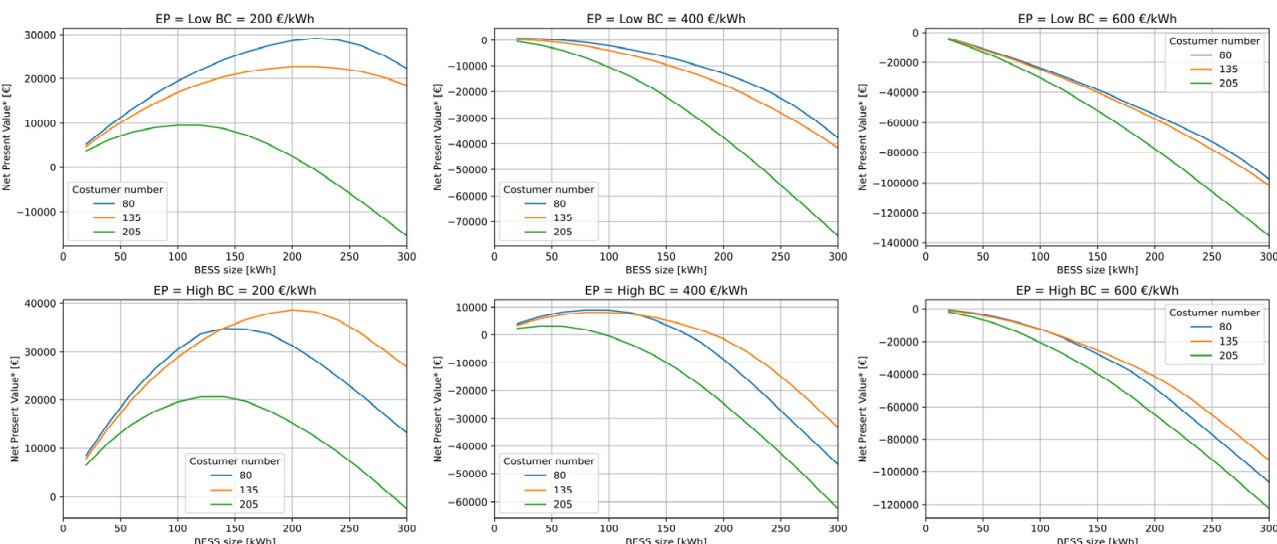

**Figure 16.** CSC: NPV* varying Energy Price (EP), BESS cost (BC), number of customers, and BESS size.

In scenarios using EA (Figure 15), a BESS cost of 600 EUR/kWh renders BESS installation economically unviable, while 400 EUR/kWh proves attractive, especially in scenarios with high energy prices. A larger BESS size is recommended for setups with a substantial surplus, while scenarios with low energy prices favor smaller BESS units. A cost of 200 EUR/kWh strikes a balance, rendering BESS integration cost-efficient across various scenarios.

Not performing EA (Figure 16) diminishes battery investment attractiveness. At 400 EUR/kWh, only smaller batteries are feasible, while 200 EUR/kWh remains appealing, even if the NPV* values achieved are lower than in the case with EA.

An intriguing observation arises regarding the impact of the customer number on battery investment. In EA scenarios, it becomes apparent that as the number of customers decreases (and REC surplus consequently increases), the NPV* of the investment rises. This trend stems from the augmented income attributed to the withdrawal by the BESS of the REC surplus energy that becomes CSC. However, in instances where EA is not implemented, this assertion holds only partially true. This is because for the BESS to be able to utilize such surplus, there must also be REC energy needs to justify its re-injection later. Consequently, it is implied that batteries should be matched to REC converging toward an equilibrium point between surplus and need, thereby optimizing both collective self-consumption (CSC) and collective self-sufficiency (CSS), i.e., with neither too many nor too few customers (Figure 16, bottom left).

Figures 17 and 18 provide further clarity by illustrating the progression of NPV, focusing on the original NPV rather than the transformed version and considering the optimal BESS size solution for each scenario. These curves offer a comprehensive perspective on investments, encompassing NPV, payback time, and battery lifetime.

In scenarios using EA (Figure 17), a BESS cost of 200 EUR/kWh presents compelling investments, ensuring a 5-year payback period in low-price scenarios and even shorter periods in high-price scenarios. A cost of 400 EUR/kWh also allows for investments with payback times of less than 10 years, albeit with more significant impacts from energy prices and customers. However, 600 EUR/kWh is evidently excessive.

Without EA (Figure 18), investments become less attractive, with payback periods extending by approximately 5 years and optimal battery sizes decreasing alongside NPV. Here, the battery cost must be around 200 EUR/kWh or less for attractiveness.

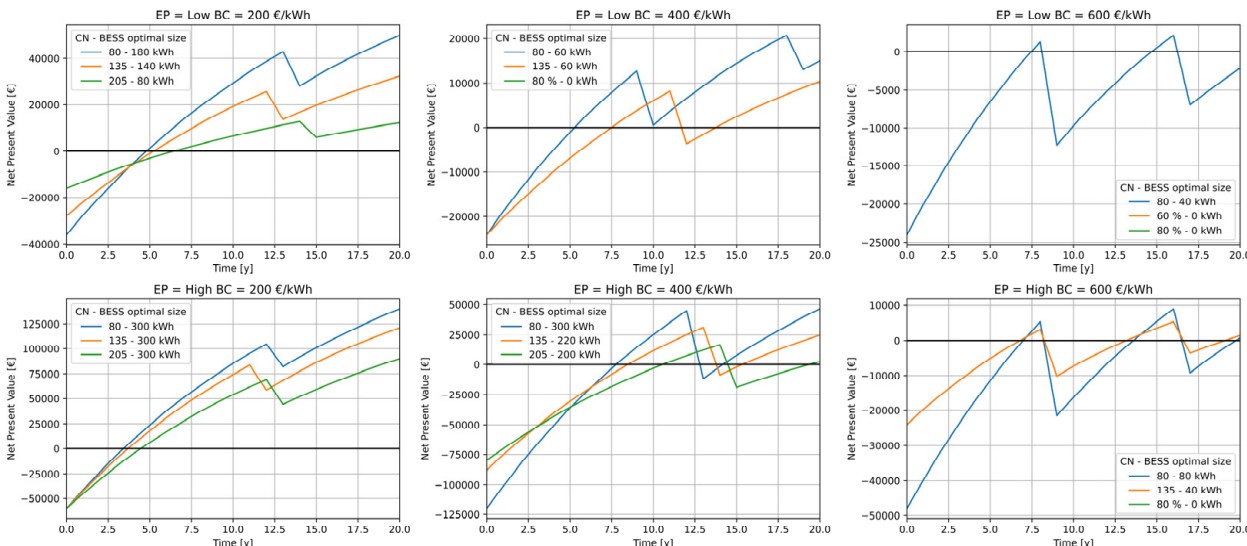

**Figure 17.** CSC + EA: optimal investments assessment for varying Energy Price (EP), BESS cost (BC), and number of customers (CN).

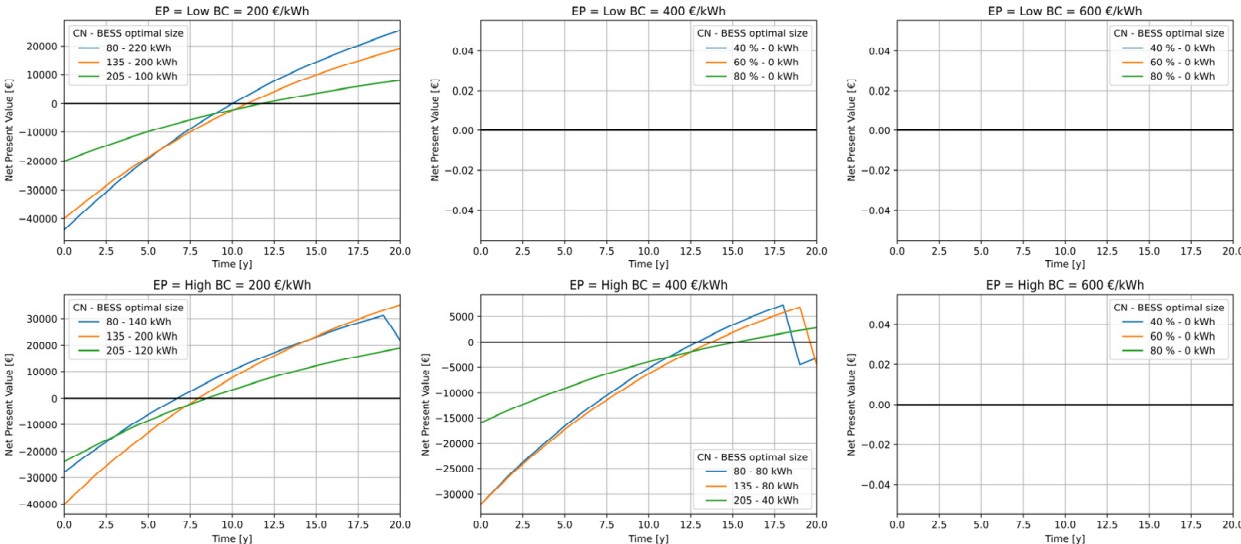

**Figure 18.** CSC: optimal investments assessment for varying Energy Price (EP), BESS cost (BC), and number of customers (CN).

Battery lifetimes exhibit steps due to replacement costs. Without EA, batteries can last about 20 years or longer; while performing EA, lifetimes vary between 7 and 13 years. Despite reduced lifetimes, the increase in cash flow, payback time, and NPV over 20 years compensates, rendering the investment more attractive overall.

## 4. Discussion

In a residential renewable energy community (REC) powered by 100 kWp of photovoltaic systems, a comprehensive techno-economic analysis was used to assess the energy and economic impact of integrating a grid-connected Battery Energy Storage System (BESS). Various scenarios were examined, accounting for factors like community size, market prices, battery characteristics, and the choice to engage in energy arbitrage (EA), with the Italian market and regulations serving as a reference.

The analysis focused on two main revenue sources: the increase in collective self-consumption (CSC) incentives resulting from BESS surplus energy withdrawal and EA, involving participation in electricity markets to capitalize on price differentials. A 20-year

simulation, considering battery aging and optimized scheduling, revealed the significance of a shared battery in enhancing collective self-consumption and sufficiency. However, scenarios with EA demonstrated higher total energy transactions compared to those without a battery.

Activation costs played a crucial role in EA scenarios, emphasizing the importance of optimizing battery scheduling and considering replacement costs. Economic findings highlighted optimal battery sizes and cost-effectiveness thresholds, with dedicated CSC batteries requiring a maximum cost of 200 EUR/kWh. Conversely, EA enabled viable investments even with costs around 400 EUR/kWh, halving the payback period and emphasizing the market's dependence on incentives.

Interestingly, the most suitable REC for BESS integration featured an intermediate number of customers, balancing surplus and demand levels unless EA was involved. In that case, REC with low customer numbers or high surpluses was preferable.

Battery aging analysis revealed that while a BESS dedicated to CSC could last 20 years, EA halved its lifespan. However, increased cash flow and net present value compensated for this reduction, rendering the investment more attractive overall.

A comparison between energy and economic optimality highlighted discrepancies, indicating the need for further reductions in battery prices, enhanced market incentives, and regulatory reviews concerning the role of grid-connected BESS within RECs.

Future studies could explore BESS potential in RECs beyond CSC and EA, considering additional revenue streams such as ancillary services. Moreover, battery management strategies should integrate real-time control for dispatching and address forecast errors and grid constraints for a comprehensive analysis.

**Author Contributions:** M.P.: Term, Conceptualization, Methodology, Software, Validation, Formal Analysis, Investigation, Data Curation, Writing—Original Draft, Visualization. L.B.: Term, Conceptualization, Methodology, Validation, Formal Analysis, Writing—Review and Editing. M.B.: Term, Conceptualization, Methodology, Validation, Formal Analysis, Writing—Review and Editing. M.I.: Term, Conceptualization, Methodology, Validation, Formal Analysis, Writing—Review and Editing. F.G.: Resources, Writing—Review and Editing, Supervision, Funding Acquisition. G.F.: Writing—Review and Editing, Supervision. C.C.: Resources, Writing—Review and Editing, Supervision, Project Administration, Funding Acquisition. All authors have read and agreed to the published version of the manuscript.

**Funding:** This research received no external funding.

**Institutional Review Board Statement:** Not applicable.

**Informed Consent Statement:** Not applicable.

**Data Availability Statement:** All the codes used are publicly available on GitHub [52,54]. Production data are downloaded from PVgis [50], and load profiles are from ARERA [51].

**Conflicts of Interest:** The authors declare no conflicts of interest.

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
