# Peer review of "Community Battery for Collective Self-Consumption and Energy Arbitrage: Independence Growth vs. Investment Cost-Effectiveness"

_sustainability, doi:10.3390/su16083111_

Round 1

Reviewer 1 Report

Comments and Suggestions for Authors

The manuscript “Community battery for Collective-Self-Consumption and Energy Arbitrage: techno-economic simulations assessing energy balances, battery ageing and different market scenarios” encompassed various scenarios considering community customers number, electricity market prices, battery cost, size and the decision to engage in energy arbitrage (EA). The Italian market and regulation are taken as a reference.

The work is interesting, while the manuscript organization is completely unacceptable for me. The style differs obviously from convenient literatures (Example: Mechanical and failure characteristics of novel tailorable architected metamaterials against crash impact, Engineering Failure Analysis, 159(2024): 108141). Therefore, this paper is suggested to be accepted only after major revisions. The following points should be addressed:

1.     The novelty and main conclusions should be summarized into a short Abstract. However, the current Abstract looks like only describing why carry out this study.

2.     Detailed literature review and brief framework should be combined as Introduction to explain why conduct this study. However, the current Introduction is much too farraginous. The novelty should be moved to Abstract, and other contents should be reorganized.

3.     All figures are blurred. The picture definition must be increased.

4.     The equation number is farraginous. For instance, Eq. (1) and Eq. (1.1) both appear.

5.     The figure number is farraginous. For instance, Fig. 2c appears and then Fig. 2a and then Fig. 2b.

6.     The discussion should be more concentrated and firm.

Comments on the Quality of English Language

Minor editing of English language is required to decrease grammar mistakes.

Reviewer 2 Report

Comments and Suggestions for Authors

Reviewer 3 Report

Comments and Suggestions for Authors

Manuscript number: sustainability-2916654

Title: Community battery for Collective-Self-Consumption and Energy Arbitrage: techno-economic simulations assessing energy balances, battery ageing and different market scenarios

The present paper is interesting. It needs some minor rectifications before one can take a final decision. The specific comments are as follows:

1)  I feel that the Title should be slightly revised to better reflect its content. I also suggest making it shorter.

2)  The paper contains some grammatical errors and typo-mistakes that should be corrected.

3)  The Abstract part should be revised. It should contain more qualitative and quantitative results.

4)  The introduction part can be further improved. A general introductory paragraph should be added at the beginning of the Introduction part on renewable energy production.

5)  Why Figures 1, 2, and 3 are repeated many times within the manuscript? The authors should check the PDF version of their article before submission.

6)  Add the (a), (b), and (c) to Figure 2.

7)  In Figure 2(b), what are the reasons for the decline in Net Present Value at a certain year?

8)  Figure 3(a) does not show a monotonic variation in the Net Present Value with the gradual increase of Activation Cost (AC) value? What are the plausible reasons for this fluctuation?

9)  How the three collective self-consumption (CSC) index, he collective self-sufficiency (CSS) index, and the incentive per costumer could be correlated together? This should be elaborated more.

10)          The style and format of references should be re-checked.

11)          The authors are advised to provide more articles, review articles, or case studies (more reliable references) as references to replace some links (weak references).

Comments on the Quality of English Language

The paper contains some grammatical errors and typo-mistakes that should be corrected.

Round 2

Reviewer 1 Report

Comments and Suggestions for Authors

 1. The organization has been improved after revisions, while the gap from most published papers still exists. The following reference is recommended to be imitated and referred. (Mechanical and failure characteristics of novel tailorable architected metamaterials against crash impact, Engineering Failure Analysis, 159(2024): 108141)

2. The style of equation number still exists many problems. For instance, Eq. 1.1 and Eq. 2 meanwhile appear.

3. The image resolution of all figures has been promoted, while some words or numbers are much too small to read.
